# Gonadal Sex Differentiation and Ovarian Organogenesis along the Cortical–Medullary Axis in Mammals

**DOI:** 10.3390/ijms232113373

**Published:** 2022-11-02

**Authors:** Kenya Imaimatsu, Aya Uchida, Ryuji Hiramatsu, Yoshiakira Kanai

**Affiliations:** 1Department of Veterinary Anatomy, The University of Tokyo, Bunkyo-ku, Tokyo 113-8654, Japan; 2RIKEN BioResouce Research Center, Tsukuba 305-0074, Japan

**Keywords:** sex differentiation, testis, ovary, folliculogenesis, cortex, medulla

## Abstract

In most mammals, the sex of the gonads is based on the fate of the supporting cell lineages, which arises from the proliferation of coelomic epithelium (CE) that surfaces on the bipotential genital ridge in both XY and XX embryos. Recent genetic studies and single-cell transcriptome analyses in mice have revealed the cellular and molecular events in the two-wave proliferation of the CE that produce the supporting cells. This proliferation contributes to the formation of the primary sex cords in the medullary region of both the testis and the ovary at the early phase of gonadal sex differentiation, as well as to that of the secondary sex cords in the cortical region of the ovary at the perinatal stage. To support gametogenesis, the testis forms seminiferous tubules in the medullary region, whereas the ovary forms follicles mainly in the cortical region. The medullary region in the ovary exhibits morphological and functional diversity among mammalian species that ranges from ovary-like to testis-like characteristics. This review focuses on the mechanism of gonadal sex differentiation along the cortical-medullary axis and compares the features of the cortical and medullary regions of the ovary in mammalian species.

## 1. Overview of Early Gonadal Supporting Cell Development in Mice

The sex of most mammals is determined by a pair of sex chromosomes at fertilization—females are XX and males are XY—whereas the genital organs of both females and males are developed from the common genital primordium. The testis in males and the ovary in females both arise from the embryonic gonad to support gametogenesis and sex hormone production. In the testis, sperm are produced in the convoluted seminiferous tubules, of which the epithelia consist of spermatogenic cells and somatic supporting cells called Sertoli cells. In the interstitium between the seminiferous tubules harbors, Leydig cells, which are the steroidogenic cells responsible for the secretion of testosterone, are present [1]. In the ovary, an egg is developed in the follicle that consists of an oocyte and a layer of somatic supporting cells called granulosa cells. Surrounding the growing follicle, a layer of theca cells, which are the steroidogenic cell lineage that produces androgen, is formed [2]. Spermatogenic cells, Sertoli cells, and Leydig cells in the testis are the counterparts of oocytes, granulosa cells, and theca cells in the ovary, respectively [3]. The fates of these cell lineages stem from the bipotential precursors in the gonadal primordium according to the sexually dimorphic genetic programs corresponding with the chromosomal sex of an individual. 

(1)
**Origin of Gonadal Supporting Cells from the Coelomic Epithelium**


In mice, the gonadal primordium (also known as the genital ridge) arises from the thickening of the coelomic epithelium (CE) on the ventral surface of the mesonephros around embryonic day 9.5 (E9.5) [4,5,6,7,8]. These bipotential gonadal primordia become evident as a pair of long and narrow structures along the anteroposterior (AP) axis by E10.5 (Figure 1A) [6,9]. The gonadal primordium develops as a result of proliferation, epithelial–mesenchymal transition, and the migration of the CE cells into the dorsal inner layers of CE via its basal lamina. Cell lineage tracing experiments indicate that CE cells are a source of somatic cell precursors, and the cells derived from the CE migrate into the medullary region, forming the primary sex cords in male and female gonads (Figure 1A) [10]. Asymmetric cell division and the ingression of CE cells into the gonad require the proper cell polarization of CE cells. In this process, the localization of NUMB (the monomeric PTB-containing adaptor protein, *Numb*), an antagonist of Notch signaling, plays an important role in establishing cell polarity in CE cells (Figure 1A) [11].

CE cells have a unique molecular signature, expressing various transcription factors including, GATA binding protein 4 (*Gata4*), orphan nuclear receptor steroidogenic factor 1 (*Sf1*, also known as *Nr5a1/Ad4bp*), and Wilm’s tumor 1 protein (*Wt1*) (Figure 1A). In mice, an anterior-to-posterior wave-like activation of GATA4 is one of the earliest molecular events in gonadogenesis; this event is essential for adequate thickening of the gonadal primordium [12]. After the onset of *Gata4* expression, *Sf1* is also expressed in an anterior-to-posterior manner [12]; sufficient expression of *Sf1* and *Wt1* in somatic progenitor cells is required for their survival [13,14,15]. The transcription factors empty spiracles homeobox 2 (*Emx2*), sine oculis homeobox 1 (*Six1*), and *Six4* also contribute to the ingression of CE cells, thickening of the gonadal primordium, and the consequent regulation of gonadal size [16,17]. In *Emx2-null* mouse mutants, CE cells exhibit severe defects in epithelial–mesenchymal transition and subsequent ingression, which result in a hypoplastic gonadal primordium [16]. *Six1/Six4* are expressed in the CE before the activation of SF1. The loss of *Six1/Six4* results in an insufficient expression of SF1, and failure of gonad expansion [17]. The loss of LIM homeobox protein 9 (*Lhx9*) also downregulates SF1 and impairs gonadal formation [18]. Therefore, SF1 is a critical transcription factor that acts downstream of these transcriptional networks during the formation and development of gonadal primordium.

Before gonadal somatic cells appear at the site of the gonadal primordium, primordial germ cells (PGCs) emerge from outside of the gonads. PGCs arise from epiblast cells in the proximal region of the early pregastrulation stages of mouse embryos around E6.5 [19,20]. PGCs subsequently move to the base of the allantois at E7.25 [21,22]; these cells migrate along the hindgut, synchronizing the morphogenic movement of the hindgut endoderm [23,24]. Next, PGCs pass through the dorsal mesentery and are finally distributed to the gonadal primordium at around E10.5 (Figure 1A) [23,25].

(2)
**SRY-Mediated Primary Sex Determination**


In the XY gonad, the gonadal primordium undergoes sex-specific differentiation soon after the arrival of PGCs [6,26,27]. The sex-determining region of the Y chromosome (*Sry*), which encodes a high-mobility group (HMG)-domain transcription factor, initiates the differentiation of supporting cell precursors into Sertoli cells within the primary sex cords [28,29]. Since its discovery, *Sry* has been regarded as a single-exon gene. However, a recent study revealed that mouse *Sry* has a cryptic second exon and the SRY protein has two distinct isoforms, the canonical SRY-S isoform and a novel SRY-T isoform [30]. SRY-S has a degron at the C-terminus, which results in its rapid degradation in vivo. In contrast, SRY-T is stably expressed in the gonads because of the absence of the degron. Furthermore, the ectopic expression of SRY-T can lead to female-to-male sex reversal in XX mice [30]. Thus, SRY-T isoform is regarded as the determinant of male fate.

Although the mechanisms underlying the upregulation of *Sry* are unclear, several signaling cascades and transcription factors—such as the mitogen-activated protein kinase (MAPK) pathway, insulin/insulin growth factor signaling, GATA4, SF1 (in humans), and WT1 —regulate *Sry* expression [15,31,32,33,34,35,36,37,38,39,40,41]. Additionally, epigenetic regulation via histone modification is important for *Sry* expression. The histone demethylase JMJD1A positively regulates *Sry* expression by controlling H3K9 methylation [42]. The histone acetyltransferases CREB-binding protein (*Crebbp*, *Cbp*, or *Kat3a*) and E1A binding protein p300 (*Ep300*, *Kat3b*, or *p300*) also regulate *Sry* expression by acetylating H3K27 at the *Sry* promoter [43].

*Sry* is transiently expressed in supporting cell precursors at E10.5–12.5. *Sry* expression is first detectable at the center of the gonad at E10.5 and extends toward cells at the anterior and posterior ends by E11.5 (Figure 1B) [44,45]. This *Sry* expression is rapidly downregulated, such that it disappears around E12.5. The loss of *Sry* causes complete XY gonadal sex reversal of the fetal and adult gonads [46]. A heat-shock-inducible SRY transgenic mouse experiment revealed that the ability of SRY to determine testis development was limited to approximately E11.0–11.25; any delay in *Sry* expression beyond E11.3 induced a sex-reversal phenotype [47]. Thus, sufficient and prompt *Sry* expression in the gonads is essential for correct testis differentiation.

The principal target of SRY is SRY-related HMG box 9 (*Sox9*), a critical testis-determining factor [48]. In XY gonads, SRY single-positive cells, SRY/SOX9 double-positive cells, and SOX9 single-positive cells are distributed in spatial order from the cortical (coelomic epithelial) side to the medullary (mesonephric) side (Figure 1B) [49,50]. The loss of *Sox9* also induces complete gonadal sex reversal. Therefore, SOX9 expression is necessary for the specification of the Sertoli cell lineage [51]. In Sertoli cell precursors, SOX9 expression induces male-specific expression of fibroblast growth factor 9 (*Fgf9*), and FGF9 promotes the high expression of SOX9. These signals form a positive-regulatory feedback loop and maintain each other’s expression (Figure 1B) [52]. FGF9 expression exhibits a center-to-pole wave-like pattern, which acts as a diffusible inducer and establishes high expression of SOX9 to induce Sertoli cell fate [53]. SOX9 expression also induces the expression of prostaglandin D2 synthase (brain) (*Ptgds)* [54]. Then, synthesized prostaglandin D2 (PGD2) from SOX9-positive Sertoli cells acts as an autocrine/paracrine factor and amplifies SOX9 signaling in a manner independent of FGF9 (Figure 1B) [54,55].

Gene expression profiling at a single-cell resolution can be used to distinguish various cell populations in an unbiased manner. The cell lineage can be predicted based on time-series sampling data via the reconstruction of developmental trajectories; these data provide insights into somatic cell differentiation during testis and ovary development. In mice, in accordance with in vivo and in vitro lineage tracing experiments, developmental trajectory analysis based on single-cell RNA sequencing (scRNA-seq) revealed that proliferating CE cells are the primary source of gonadal somatic cells for both XY and XX gonads [56,57]. A study based on scRNA-seq showed that a new uncharacterized somatic cell population, supporting-like cells (SLCs), contributes to the formation of rete testis and rete ovarii [58]. Rete testis is an anastomosing canal connecting the seminiferous tubules and the epididymis [59]; rete ovarii is a group of anastomosing tubules located in the hilum of the ovary [60]. SLC lineage is the first distinct somatic cell lineage specified in the bipotential gonads, as early as E10.5, before the initiation of gonadal sex determination. SLC progenitors initially express both *Wnt4* and *Sox9*, and they become sexually dimorphic around E12.5. Later, gene expression in SLCs becomes more Sertoli-like or granulosa-like state and mainly contributes to the formation of rete testis or rete ovarii, respectively [58].

Testicular cords, which develop into the seminiferous tubules in the mature testis, are formed via the aggregation of Sertoli cells and germ cells in the medullary region of XY gonads by E12.5 (Figure 1C). After E11.5, the proliferation of CE cells is induced in response to platelet-derived growth factor (PDGF) in XY gonads. The cells proliferating during E11.5–E13.5 only differentiate into interstitial cells, resulting in a rapid increase in testis size [61]. During the same period, the testis-specific formation of the tunica albuginea, the fibrous capsule of the testis, and the vasculature separate the CE layer from the testicular parenchyma [3]. This possibly causes the cessation of the ingression toward the testicular parenchyma and represses the further expansion of the cortical region.

Differentiated Sertoli cells produce various autocrine/paracrine signals, which induce the differentiation of other testicular cell populations. For this reason, the proper establishment of Sertoli cells depending on the expression of *Sry* and its downstream gene cascade is thought to be a crucial event in sex differentiation in the testis. Anti-Müllerian hormone (AMH), a paracrine factor secreted by SOX9-positive supporting cells, induces the regression of the Müllerian duct, a primordium of the female reproductive tract (Figure 1C) [62,63]. The loss of functional AMH or its receptor AMHR2 results in males with a female reproductive tract derived from the Müllerian duct in mice and humans with a fully virilized phenotype; this condition is known as Persistent Müllerian duct syndrome (PMDS) [63,64,65,66,67]. AMH secreted by Sertoli cells decreases the size of the Leydig cell population [63]. Desert hedgehog (DHH) is a paracrine factor secreted by differentiated Sertoli cells; the expression of DHH is induced by the expression of SRY and SOX9 [68,69]. DHH binds to its receptor PTCH1 in the interstitial progenitor cell population and upregulates gene expression by activating its downstream transcription factor, GLI1 (Figure 1C). DHH and its signaling cascade upregulates SF1 and cytochrome P450, family 11, subfamily a, polypeptide 1 (*Cyp11a1*), which are steroidogenic enzymes necessary for androgen synthesis [68,70]. Thus, DHH produced in Sertoli cells induces the specification of fetal steroidogenic Leydig cells. In mice, fetal Leydig cells secrete androstenedione and other androgens (i.e., precursors of testosterone), which are used by Sertoli cells to produce testosterone (Figure 1C) [71,72]. Masculinization of the internal and external genital tract, known as secondary sex determination, is promoted by testosterone from Sertoli cells in differentiated testes. This process is also promoted by 5a-dihydrotestosterone, which is derived from testosterone by 5a-reductase in the target tissues [73].

Taken together, the molecular and cellular pathway of the SRY-dependent testis determination in XY gonads is well-defined, and the formation of the functional testicular components mainly occurs in the medullary region.

## 2. Molecular and Cellular Events in Ovarian Somatic Cells

(1)
**Female Fate Determination in Somatic Supporting Cells in the Early Phase of Ovarian Development**


At the early phase of ovarian development, granulosa cells are differentiated from common somatic precursor cells with Sertoli cells in the gonadal primordium [44]. Around E10, CE cells covering the gonadal primordium proliferate and ingress into the gonadal parenchyma at the ventral side of the mesonephric region, leading to the formation of a primary sex cord (Figure 2A) [3,10]. Wingless-type MMTV integration site family member 4 (WNT4) regulates the thickening of the CE and its expansion toward the subepithelial region in both XX and XY gonads [74]. In conjunction with R-spondin homolog 1 (RSPO1), a ligand of the leucine-rich repeat-containing G-protein-coupled receptors LGR4 and LGR5 regulate the WNT4/β-catenin signaling pathway [75]. At E11.25–11.5, *Wnt4* and *Rspo1* are expressed, and β-catenin signaling is activated in the coelomic region of both XX and XY gonads. Thus, RSPO1/WNT4/β-catenin signaling does not exhibit sexual dimorphism at the beginning of the gonadal sex differentiation process [74]. Around E11.5, 12 h after the initiation of *Sry* expression in the central region of the XY gonads [45,50], the XX gonadal primordium promotes ovarian differentiation by inducing pro-ovarian genes and repressing pro-testis genes [76,77,78]. From E11.5, the activation of RSPO1/WNT4/β-catenin signaling in XX gonads promotes the expression of pro-ovarian genes such as the X-linked nuclear receptor *Nr0b1*/*Dax1* [79], follistatin (*Fst* [80]), Tgfb-related genes, bone morphogenetic protein 2 (*Bmp2* [80]) and the homeobox gene *Irx3* [81]. RSPO1/WNT4/β-catenin signaling in the ovary also promotes the proliferation and survival of oogonia, as well as the meiotic initiation of female germ cells [82,83]. Furthermore, RSPO1/WNT4/β-catenin signaling mediates ovarian development by antagonizing the masculinizing factors SOX9 and FGF9, which are downstream of SRY [52,53,84]. XX gonads in *Wnt4* or *Rspo1* mutant mice show partial sex reversal, including the upregulation of *Sox9* and the appearance of testicular vasculature [85,86,87,88,89]. In contrast, the stabilization of β-catenin signaling in XY gonads induces partial male-to-female sex reversal [90]. Furthermore, heat-induced ectopic expression of *Sry* via the Hsp70.3 promoter in XX gonads between E11.0 and 11.25, when *Sry* expression is initiated in the central region of XY gonads, can lead to stable induction of SOX9 expression and testicular cord formation. However, ectopic expression in XX gonads at E11.3–11.75, beyond the window of *Sry* action, induces SOX9 expression but fails to stabilize SOX9 expression or form testicular cords. This time window can be prolonged by reducing WNT4 activity, which rescues the stable expression of SOX9 and the subsequent formation of testicular cords [47]. Therefore, WNT4 signaling promotes ovarian development by antagonizing the stabilization of SOX9 expression at the early phase of gonadal sex differentiation. In contrast, ectopic expression of *Sry* after E12.0 induces SOX9 expression in only a few somatic cells in the medullary region of XX gonads; the reduction in WNT4 activity cannot rescue SOX9 induction after E12.0 [91]. This result indicates that the robustness of the ovarian program in the supporting cells of XX gonads is established around E12.0. Additionally, it also suggests that WNT4 signaling promotes ovarian development by repressing the stabilization of SOX9 expression rather than by repressing SOX9 expression in the gonadal supporting cells.

In mouse XX gonads, the ovarian-specific transcriptional program begins around E11.5 [77,92,93,94]. Forkhead box L2 (FOXL2) is an early ovarian factor that is essential for the fate and phenotype of granulosa cells [95,96,97,98]. FOXL2 expression is induced later than the activation of the WNT4/β-catenin pathway [77]; the loss of *Ctnnb1* (β-catenin) decreases FOXL2 expression, whereas the stabilization of β-catenin in XY gonads induces FOXL2 expression [90]. These results suggest that RSPO1/WNT4/β-catenin signaling is essential for FOXL2 upregulation in granulosa cells. FOXL2 is expressed in both granulosa cells and theca cells (i.e., ovarian steroidogenic cell lineages), and it has multiple roles in ovarian development. In granulosa cells, FOXL2 represses the expression of SOX9 and DMRT1, which are evolutionally conserved DM-domain transcriptional factors that induce testis differentiation by repressing female reprogramming in postnatal mammalian testis [99,100,101]. FOXL2 expression in mouse XX gonads is first observed in gonadal somatic cells in the center-medullary region near the adjacent mesonephros around E12.0; this expression rapidly expands throughout the ovarian parenchyma by E12.5 [91]. After E12.5, most FOXL2-positive somatic cells are located in the medullary region, whereas a few of them are located within the coelomic domain (Figure 2B) [102]. The medullary granulosa cells undergo mitotic arrest upon the upregulation of the cyclin-dependent kinase inhibitor p27 [103]. The maintenance of p27 in XX somatic cells requires FOXL2 [103,104], but FOXL2 is not required for mitotic arrest in XX granulosa cells [103], suggesting that another factor is involved in mitotic arrest. These findings indicate a balance between supporting cell self-renewal and differentiation in the developing ovary: these processes are maintained by WNT4/β-catenin and FOXL2/p27, respectively [103].

There are functional differences between mice and other mammalian species in relation to the role of *Foxl2* and related genes in ovarian development. In goats, FOXL2 is a key female sex-determining gene [105,106]. *FOXL2* is a gene responsible for polled intersex syndrome (PIS), which leads to XX female-to-male sex reversal associated with the absence of horn growth [106]. In the fetal XX gonads of *FOXL2* knock-out goats, *SOX9* and *DMRT1* are upregulated in Sertoli-like cells, forming testicular cords [105]. In mice, in contrast, the loss of *Foxl2* in XX gonads does not induce an appreciable sex-reversal phenotype at the fetal stage and birth, although it results in the transdifferentiation of granulosa cells into Sertoli-like cells in the postnatal stage [96,98,107]. These results indicate that FOXL2 is not necessary for the initiation of granulosa cell differentiation, but it has a crucial role in the maintenance of granulosa cell identity in mice. It has been reported that the compensatory/redundancy functions of RUNX1 with FOXL2 have a role in granulosa cell differentiation of the mouse fetal ovary [108]. RUNX1 is expressed in the supporting cell lineages of both XX and XY gonads at E11.5, and subsequently shows ovary-biased expression in the pre-granulosa cells at E12.5. The loss of *Runx1* alone in the somatic cells of the XX ovary results in normal ovarian morphogenesis even in the postnatal stage. However, *Runx1*/*Foxl2* double-knock-out mice show the partial masculinized phenotype of the XX ovary around E15.5. At birth, the XX gonads in *Runx1*/*Foxl2* double-knock-out mice form testis cord-like structures with DMRT1-positive cells surrounding clusters of germ cells, although SOX9 expression is not observed [108]. RSPO1/WNT4/β-catenin signaling also has a compensatory function along with FOXL2 during ovarian differentiation. In *Wnt4*/*Foxl2* double-knock-out mice, the XX gonads at birth show a masculinized phenotype, including the upregulation of *Sox9* and *Dmrt1* expression in the supporting cells, forming testicular cords [97]. The XX gonads of *Rspo1*/*Foxl2* double-knock-out mice around birth also show ovotestis-like structures composed of both ovarian and testicular parts [109]. Furthermore, FOXL2 upregulates aromatase factors including CYP19A1, an enzyme responsible for a key step in the biosynthesis of estrogens, in a female-specific manner in theca cells [110,111,112]. Additionally, FOXL2 maintains the identity of granulosa cells in the adult ovary by repressing SOX9 expression through the mobilization of estrogen signaling [113]. The loss of estrogen receptor alpha and beta (ESR1 and ESR2) induces transdifferentiation of the mutant ovary, including Sertoli-like cells that express AMH and SOX9 [114]. This evidence indicates that the cooperative activity of FOXL2, RUNX1, RSPO1/WNT4/β-catenin signaling, and estrogen signaling is necessary for granulosa cell fate in the mouse ovary. However, high expression of *RUNX1* in the fetal ovary has been detected in goats and humans [108], and FOXL2 regulates the expression of *RSPO1* and *CYP19A1* in the goat ovary [100,105]. These suggest that the key player genes in ovarian development are conserved among various mammalian species, but the functional difference concerning FOXL2 and related factors in relation to granulosa cell differentiation between mice and goats are hard to explain simply via compensatory and synergistic interaction. The comparison of gene regulatory networks in granulosa cell differentiation among mammalian species may provide us with a further understanding of the developmental pathway of the mammalian ovary.

(2)
**Secondary Population of Granulosa Cells in the Cortical Region of the Ovary**


In contrast to mouse XY gonads, the CE in XX gonads continuously exhibits proliferation, ingression, and expansion [102]. This contributes to the formation of ovigerous cords that consist of female germ cells and the surrounding pregranulosa cells by the perinatal stage. These cords are regarded as secondary sex cords or ovarian cords (Figure 2B).

In the fetal to adult stages, LGR4 and LGR5, receptors for RSPO1, are expressed in the cortical region of XX gonads, including the proliferative region of the ovarian surface epithelium (Figure 2B) [115,116]; the expression of LGR5 after E12.5 is dependent on RSPO1/WNT4/β-catenin signaling [116]. LGR4 and LGR5 are markers for tissue stem cells, such as stem cells in the mammary gland, intestine, and hair follicle [117,118,119,120]. LGR4 and LGR5 are also markers of stem/progenitor cells in the ovarian surface epithelium, which generate new granulosa cells that contribute to cortical follicles until birth (Figure 2B) [116,121]. In the postnatal ovary after the cessation of granulosa cell recruitment, *Lgr5* is restricted to stem cells in the ovarian surface epithelium; moreover, *Lgr5* promotes the regenerative repair of ovulatory wounds in the adult ovary [115]. In both XX and XY gonads, RSPO1/WNT4/β-catenin signaling is involved in the proliferation of the CE in the early stage of gonadogenesis [74]. In contrast, the expression levels of LGR4 and LGR5 are higher in XX gonads compared with XY gonads after E12.5; this timing coincides with the ovarian-specific proliferation of the CE throughout the fetal and perinatal stages (Figure 2B,C). Notably, *Lgr5* expression is downregulated in medullary granulosa cells [103]. The presence of LGR5 and the presence of FOXL2 are mutually exclusive; the expression patterns of these markers exhibit a gradient from the cortex (LGR5-positive and FOXL2-negative rich) and the medulla (FOXL2-positive and LGR5-negative rich) in somatic cells of the fetal ovary [116]. However, LGR5-positive cortical cells subsequently differentiate in response to unknown signals that initiate the expression of p27 and FOXL2 from the perinatal stage [103].

An scRNA-seq analysis also demonstrated the heterogeneity and origins of two distinct lineages of ovarian supporting cells characterized by their unique transcriptome [122]. Granulosa cells can be categorized into two populations. The first population is bipotential-derived pregranulosa cells (BPG cells) characterized by *Wnt6* expression and distinguishable as early as E11.5, as a counterpart of Sertoli cell lineage in XY gonads. The second population is epithelial-derived pregranulosa cells (EPG cells), which can be distinguished at E12.5; these cells are labeled with *Lgr5* expression as early as E13.5. In the E14.5 ovary, EPG cells express both *Lgr5* and *Gng13* (guanine nucleotide-binding protein (G protein], gamma 13); the expression of *Gng13* is restricted to the ovarian surface during sexual differentiation [123] and distributed around cysts in the outer ovarian cortex. The generation of pregranulosa cells from the ovarian surface epithelium has been suggested to be limited to the fetal gonad, and cortical BPG cells are entirely displaced by EPG cells in mature ovaries [122].

These findings indicate the presence of two distinct populations of ovarian supporting cells derived from the CE. Medullary supporting cells are differentiated from the same progenitor, along with testis Sertoli cells, at the early phase of sex differentiation and cortical supporting cells arise from LGR5-positive CE cells throughout the fetal and perinatal stages.

(3)
**Cortical–Medullary Regionality of Folliculogenesis Waves**


In mice, PGCs move to the gonadal primordium around E10.5, where they undergo active proliferation, which leads to the formation of germ cell cysts by E14.5 [124,125]. One germ cell produces approximately 30 cell clones via proliferation, which results in the formation of an average of 4.8 cysts that comprise five or six germ cells connected by an intercellular bridge [125]. This cyst formation occurs homogenously throughout the ovarian parenchyma, and the cysts are organized into ovigerous cords along the cortical–medullary axis by E14.5 [125]. Beginning at E12.5, meiosis in female germ cells in mouse XX gonads is initiated by retinoic acid signaling from anterior mesonephric tissue (Figure 2B) [126,127], which is activated by stimulated by retinoic acid 8 (*Stra8*) and REC8 meiotic recombination protein (*Rec8*) in an anterior-to-posterior manner [128,129].

In mouse XX gonads, the cyst breakdown of the interconnected germ cells begins in the medullary region, followed by the cortical region from E14.5 to E17.5 [125,130]; the cysts then become single or double-connected germ cells. Furthermore, some oocytes are eliminated via apoptosis, mainly in the medullary region, which results in the enrichment of germ cells in the cortical region [131,132,133]. In the perinatal stage, a surviving germ cell is surrounded by a layer of squamous somatic supporting cells, which results in the formation of a primordial follicle. Primordial follicles undergo folliculogenesis, ending with either ovulation or follicular death (i.e., atresia). The stages of folliculogenesis are (1) primary follicles with a single layer of granulosa cells surrounding the oogonia, (2) secondary follicles consisting of a stratified epithelium of granulosa cells surrounding the oogonia, and (3) tertiary follicles with a well-developed central cavity called an antrum. In mice, the first wave of folliculogenesis occurs in the medullary region soon after birth, and the pregranulosa cells in the medullary region contribute to the first follicles (Figure 2C) [102,121]. Subsequently, a wave of folliculogenesis occurs in the cortical region, which harbors the granulosa cells that are newly recruited from stem/progenitor cells in the LGR5-positive ovarian surface epithelium (Figure 2C) [116,121]. In tertiary follicles, the layer of theca cells surrounding the follicle is well-defined. The stromal progenitor cells, which express nuclear receptor subfamily 2 group F member 2 (NR2F2) [116] (Figure 2B) and contribute to theca cells, originate from the CE along with supporting progenitor cells. Nevertheless, the mechanism underlying the diversification of these lineages remains unknown. In the mouse ovary, granulosa cells produce DHH and Indian hedgehog (IHH), whereas theca cell progenitors express PTCH1 and GLI1 (the receptors for hedgehog signaling) around birth [134,135]. Additionally, the GLI1-positive cell population migrating from the mesonephros contributes to theca cell progenitors [134]. The molecular mechanism of theca cell differentiation indicates that steroidogenic cell differentiation in both the testis and ovary share a common signaling pathway.

## 3. Diversity of Ovarian Organogenesis along the Cortical–Medullary Axis

In the testis developmental pathway, the male-specific gene cascade (SRY, SOX9, and AMH) is highly conserved among mammals. In contrast, ovarian organogenesis—such as the formation of secondary sex cords, the initiation of meiosis in germ cells, and the timing of folliculogenesis—exhibits considerable diversity among mammalian species. To understand this diversity, it may be helpful to focus on the regionalization of the medullary region (the primary sex cords) and the cortical region (the secondary sex cords), which correspond to the region of future folliculogenesis in the ovary (Figure 3A). Indeed, the morphological characteristics of the ovarian medullary region exhibit considerable diversity among mammal species.

In the ovaries of goats and cows, species with a longer gestating period (*Capra hircus*, 150 days; *Bos taurus*, 280 days), distinct secondary sex cords in the cortical region are formed during the fetal period; folliculogenesis occurs before birth. In contrast, the few germ cells in the medullary region immediately disappear during the fetal period, and the primary sex cords regress (Figure 3B) [136]. In the ovary of mice, a species with a short gestation period (20 days), primary sex cords are formed in the medullary region, whereas secondary sex cords in the cortical region are not distinct during the fetal period. In the ovaries of mice, unlike other mammals, most oocytes in primary sex cords are maintained and contribute to follicle development immediately after birth as the first wave of folliculogenesis (Figure 3B), whereas the follicles in the medullary region are regressed after the first wave [137]. In contrast, primordial follicles in the cortical region developed continuously according to the estrous cycle after sexual maturity.

Horses (*Equus caballus*) have large developed medullary region in the ovary. The equine ovary has a unique structure that includes a concave cortical region known as the ovulation fossa [2,136]. The cortical region is restricted to the central area enclosed within a dense, richly vascularized connective tissue casing, which corresponds to medulla [138]; folliculogenesis is limited to the central cortical region (Figure 3B).

In some mammalian species, ovarian development exhibits a unique pattern, with an ovotestis-like structure containing ovarian tissue in the cortical region and a testis-like structure in the medullary region. In the ovary of the spotted hyena (*Crocuta crocuta*; gestation period, 110 days), the medullary region is separated from the cortical region by a connective tissue boundary during the mid-gestation period. In the medullary region, a cluster of cells expresses 3βHSD; these cells are regarded as Leydig cell-like steroidogenic cells (Figure 3B) [139]. In contrast, AMH expression is not present in the cortical or medullary region, suggesting that the supporting cells are not masculinized [139]. In the ovaries of most species of moles, secondary sex cords are formed earlier than the testicular cords in male moles [140]. Additionally, a large medullary region develops that encompasses Leydig-like cells and a testicular cord-like structure [140]; this structure does not exhibit the expression of SOX9 or AMH (Figure 3B) [141]. Notably, the normal female spotted hyena has unique external genitalia, comprising a large pendulous penis-like clitoris with a urogenital sinus continuing to the vagina [142,143]; the masculinization of the female external genitalia is largely independent on the influence of androgens [144,145]. The female European mole (*Talpa europaea*) has a similar penis-like clitoris [145], but the role of androgens in its development is unclear.

The ovarian cortical–medullary pattern is also present in non-mammalian vertebrates. In chicks, sex differentiation depends on the ZZ/ZW system. In the ZW female embryo, gonads are asymmetrically differentiated. The left gonad develops into a functional ovary, which contains a cortical region with oocytes, as well as the medullary cords with lacuna. In contrast, the right gonad, which has a medullary cord and undifferentiated cortex, regresses [146,147,148]. In reptiles, the mechanism of sex differentiation in the red-eared slider turtle (*Trachemys scripta*) has been well-studied. Similar to other reptiles, sex differentiation in *T. scripta* is dependent on temperature; if eggs at stage 16 or earlier incubate at 26 ºC, nearly 100% of eggs become males, and if eggs at or before stage 17 incubate at 31 ºC, nearly all eggs become females [149,150]. Thus, reptiles have a different mechanism of gonadal sex differentiation to that of mammals and birds. In contrast, primary sex cords continuous with the CE are present in both male and female gonads. In the male gonads, primary sex cords become testis cords in the medullary region, whereas the cords form lacunae in the medullary region of the female gonads [151]. This temperature-dependent sex differentiation in *T. scripta* embryo is regulated by the activation of the histone H3 lysine 27 (H3K27) demethylase KDM6B, which promotes the transcription of the male sex-determining gene *Dmrt1* [152]. At 31 ºC, intracellular calcium increases and the levels of phosphorylated STAT3 is elevated at stage 15 and 17, and becomes mostly restricted to the cortical region by stage 20. The phosphorylated STAT3 repress *Kdm6b* transcription, blocking the induction of *Dmrt1* and the male pathway [153]. Taken together, the regionalization of the gonad along the cortical–medullary axis is conserved among various vertebrate species, even when different mechanisms of gonadal sex differentiation are adopted.

These comparative findings regarding mammalian ovary development indicate that the differentiation of the medullary region, which corresponds to the primary sex cords of the fetal ovary, exhibits morphological and functional diversity that ranges from an ovary-like structure to a testis-like structure. Some reports suggest that the sex of the primary sex cords and the medullary region is more inclined towards the male phenotype. AMH is expressed in the primary sex cords in the XX gonad, but they are absent from the secondary sex cords that correspond to the cortical region in the differentiated ovary [154]. The transient and forced expression of SRY in the ovary using heat-shock-inducible SRY in transgenic mouse experiments [47] led to SOX9 expression in granulosa cells of the first-wave follicles in the medullary region [91]. In contrast, forced SRY induction could not induce SOX9-positive cells in the medullary region of the ovaries of XO female mice, in which first-wave follicles were absent [154]. These data indicate the presence of male-inclined features of the primary sex cords and the medullary region in the ovary and raise the possibility that these features cause or contribute to abnormal masculinization in XX disorders of sexual differentiation in mammals.

## 4. Conclusions

In this review, we have described recent findings regarding the molecular and cellular events of the somatic cell lineage in gonadal sexual differentiation and ovarian organogenesis. We have also compared the ovarian morphological characteristics of mammalian species, with a focus on features in the cortical and medullary regions of the gonads. Regionalization along the cortical–medullary axis of the gonads is well-known in various vertebrate species. In 1950s, Witschi proposed that the medulla has the potential to derive testicular tissues and that the cortex is the precursor of ovarian tissue by the morphological observation in non-mammalian vertebrate species [155]. In contrast, the findings obtained using mouse models with XX-XY chimeras, cell tracing and lineage analysis have demonstrated the presence of common precursors for Sertoli cells and granulosa cells arising from the CE [10,44,156,157]. This evidence seems to contradict Witshi’s regionalization model and indicates that the CE is the common origin of the supporting cell lineage in both XX and XY gonads. Furthermore, recent transcriptome analysis and genetic studies have revealed that the two-step-proliferation of the CE results in primary sex cord formation in both XX and XY gonads at the early phase of gonadal sexual differentiation, and secondary sex cord formation in XX gonads by the prenatal stage. The two-step formation of the sex cords is coincident not only with Witshi’s regionalization model but also with the common precursors for the supporting cells of both sexes from the CE. The primary sex cords in both XX and XY gonads contribute to the supporting cell lineage in the medullary region, and only the secondary sex cords in the XX gonads contribute to that in the cortical region. However, research has not revealed why the formation of the secondary sex cords occurs only in the XX gonad, despite the primary sex cords being formed in both XX and XY gonads. The expression of LGR5, the key factor in the formation of secondary sex cords, is enriched in the XX gonads from E12.5 depending on RSPO1/WNT4/β-catenin signaling [103,116]. RSPO1/WNT4/β-catenin signaling is activated in the XX gonads after E11.5 [76,77,78]. Therefore, the formation of secondary sex cords can be considered as an event that is part of the ovarian developmental program. However, canonical β-catenin signaling is activated in the coelomic epithelium of both XX and XY gonads at E11.5 [74]. Furthermore, the expression of *Lgr5* has been detected even in the XY gonads at E12.5, although the expression level in the XY gonad is significantly lower than that in the XX gonads [116]. Interestingly, the upregulation of *Lgr5* in the XY gonads is transient, and its expression becomes weak at E13.5 [116]. Considering these expression patterns, there is a possibility that the proliferation of the CE cells contributing to secondary sex cords is inhibited in the testis developmental pathway. In testis development, *Sry* triggers Sertoli cell differentiation and paracrine/autocrine factors secreted from Sertoli cells play vital roles in the differentiation of the other testicular cells and the formation of a testis-specific structure. In fact, the proliferation of CE cells after E11.5 is induced in response to platelet-derived growth factor (PDGF) secreted from Sertoli cells, but the cells proliferating during E11.5-E13.5 only differentiate into interstitial cells, not into supporting cells [61]. Together with the testis-specific formation of the vasculature and tunica albuginea beneath the CE, the Sry-downstream pathway may maintain the robustness of testis development by inhibiting secondary sex cord formation and granulosa cell occurrence through the paracrine factors secreted from Sertoli cells. Further analysis regarding the sex dimorphic regulatory mechanism of sex cord formation will contribute to understanding how to organize the proper regionalization of the gonad in each sex.

Components of gametogenesis are mainly formed in the medullary region in the testis and the cortical region in the ovary; this pattern is conserved among non-mammalian vertebrates, such as chicks and turtles [147,148,151,155]. In contrast, the fate determination of the ovarian medulla, which corresponds to the primary sex cord of the fetal ovary in mammals, exhibits morphological diversity that ranges from folliculogenesis to showing testis-like characteristics, such as the presence of Leydig-like cells (Figure 3). The function of this diversity and plasticity of the medullary region of the ovary in mammalian species has not been determined. In the spotted hyena, the aggressive and dominant behavior of adult females is mitigated by suppressing their androgen levels, resulting in less intense or less prolonged behavior [158,159]. Additionally, female moles show aggressive behavior in defending their territories, similar to that of males, suggesting the effect of testosterone secreted from the medullary region on the ovary [160]. Therefore, masculinization in the medullary region of the ovaries of spotted hyenas and moles is evolutionally promoted to enhance the acquisition of aggressive traits in females. The diversity of the ovarian medulla may be important for their reproduction, as well as behavioral and survival strategies. The plasticity of the medullary region of the ovary may be utilized for environmental adaptation.

## Figures and Tables

**Figure 1 ijms-23-13373-f001:**
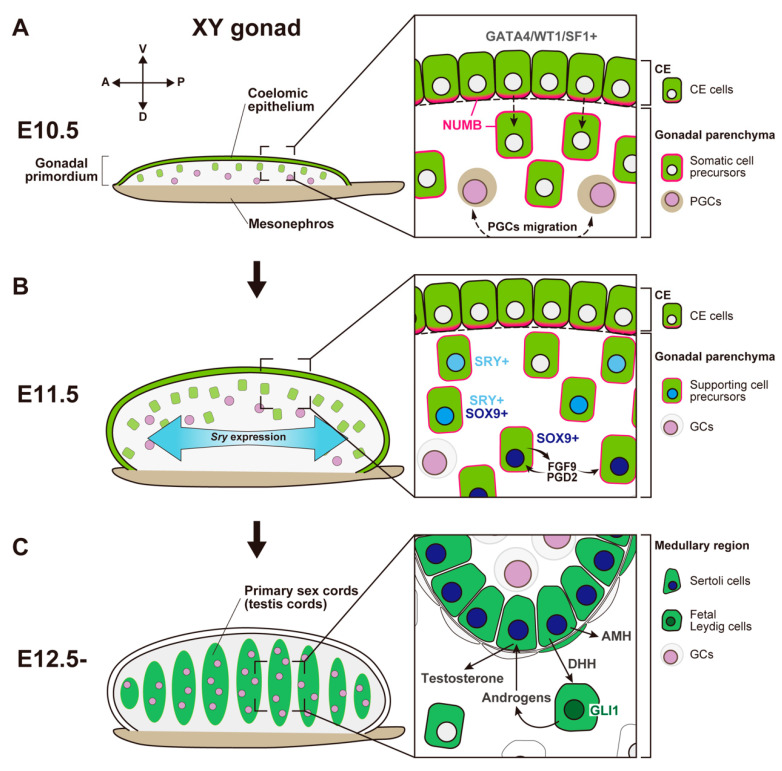
Schematic representation showing morphological change and somatic cell differentiation during the sex determination period, from the bipotential gonadal primordium at E10.5 to the differentiated testis at E12.5. (**A**) At E10.5, the gonadal primordium appears as a long and narrow structure composed of the coelomic epithelium (CE) and the migrated CE cells (somatic cell precursors). CE cells express GATA4/WT1/SF1, and their ingression and asymmetric cell division are primarily controlled by Notch signaling with the NUMB distribution (magenta) and Notch signaling. PGCs migration occurs around E10.5. (**B**) By E11.5, the gonadal primordium expands along the dorsoventral axis via the ingression of CE cells. Testis-specific *Sry* expression occurs in a center-to-pole manner along the anteroposterior axis. Beneath the CE, supporting cell precursors, SRY single- (cyan nuclei), SRY/SOX9 double- (blue nuclei), and SOX9 single- (deep-blue nuclei) positive cells, are distributed from the CE (dorsal)-to-mesonephric (ventral) side. SOX9-positive Sertoli cells secrete FGF9 and PGD2, and these factors upregulate and maintain *Sox9* expression in the own and neighboring supporting cell precursors. (**C**) At E12.5, testis cords are formed by Sertoli cells and germ cells (GCs). SOX9-positive Sertoli cells secrete paracrine factors, such as AMH and DHH. DHH signaling activates its downstream factor, GLI1, in fetal Leydig cell progenitors. Fetal Leydig cells with activated GLI1 (deep-green nuclei) produce androgens, which are converted to testosterone by Sertoli cells and then induce proper differentiation of the internal and external genital tract.

**Figure 2 ijms-23-13373-f002:**
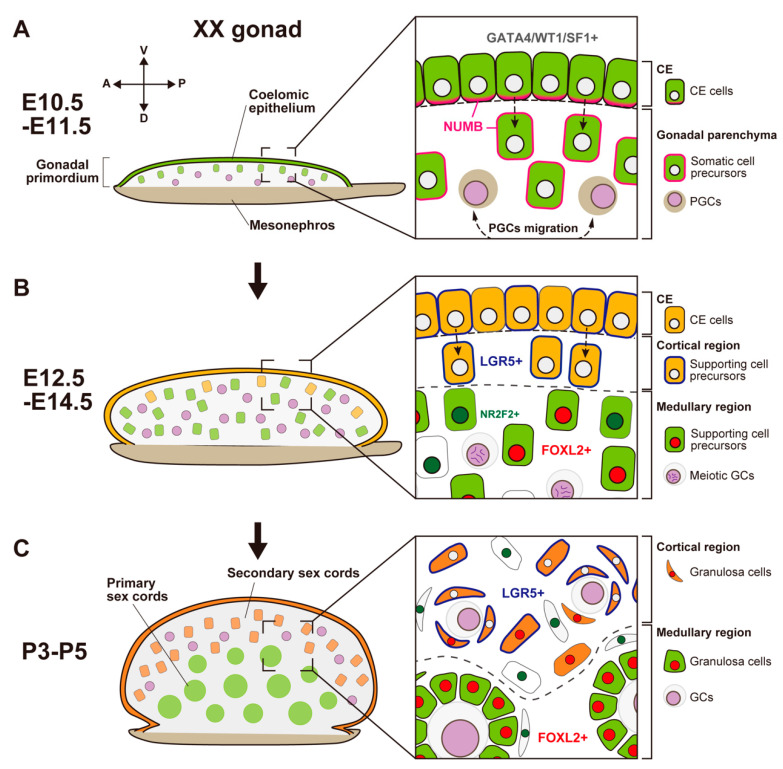
Schematic representation showing morphological changes and somatic cell differentiation along the cortical-medullary axis during fetal and postnatal stages, from the bipotential gonadal primordium at E10.5 to the differentiated ovary around P5. (**A**) At E10.5, the CE cells proliferate and ingress into the gonadal parenchyma, controlled by the NUMB distribution (magenta) and Notch signaling, to form primary sex cords without sexual dimorphism. (**B**) At E12.5, supporting cell precursors in the medullary region (green cells) express an early ovarian factor, FOXL2 (red). In contrast to testis differentiation, the proliferation and ingression of the CE in XX gonads continues after E12.5, leading to the formation of secondary sex cords in the cortical region (orange cells). Supporting cell precursors in secondary sex cords, including CE cells, express the transmembrane receptor LGR5 (blue). LGR5 and FOXL2 expression is mutually exclusive, and it exhibits a gradient from the cortex (CE side) to the medulla (mesonephric side). Other somatic cells (i.e., interstitial precursor cells) also exclusively express NR2F2 (deep green). Germ cells (GCs) initiate meiosis at E12.5 (deep purple, condensed chromosomes). (**C**) At postnatal stages, FOXL2-positive granulosa cells in primary sex cords in the medullary region contribute to the formation of the first wave of follicles. In the cortical region, granulosa cells that originate from LGR5-positive cells (orange cells) form the secondary sex cords and primordial follicle pools. After sexual maturity, these follicles develop according to the estrous cycle.

**Figure 3 ijms-23-13373-f003:**
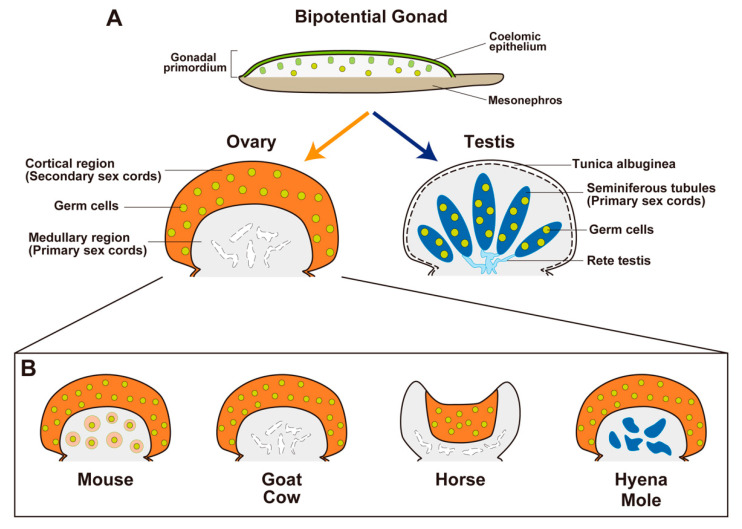
Sexual differentiation of bipotential gonads and the histological variations of ovaries in the indicated species. (**A**) Bipotential gonads differentiate into testes or ovaries. In mammalian testes, primary sex cords form tubular structures (seminiferous tubules (deep blue) with germ cells (yellow dots)) and the rete testis (cyan), which provide a route for continuous sperm transportation from the seminiferous tubules to the efferent ducts. Interstitial components are shown in gray. Mammalian ovaries commonly develop secondary sex cords with germ cells in the cortical region as a site of future folliculogenesis (orange; yellow dots, germ cells). However, the morphological characteristics of primary sex cords in the medullary region of the ovary show considerable diversity among mammals. (**B**) In the ovaries of goats and cows, secondary sex cords in the cortical region form during the fetal period. Germ cells in the medullary region disappear and the primary sex region regresses. In the ovaries of mice, secondary sex cords in the cortical region are not distinct in the fetal period; germ cells in the medullary region are maintained and develop immediately after birth as the first wave of follicles (pale-orange circles with yellow dots). In the ovaries of spotted hyenas and most mole species, the medullary region develops as a male-like tissue with Leydig-cell-like steroidogenic cells (deep blue). In the ovaries of horses, the cortical region (in which folliculogenesis occurs) is surrounded by the well-developed medulla and forms a unique structure known as the ovulation fossa.

## Data Availability

Not applicable.

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
