# Peer review of "Gonadal Sex Differentiation and Ovarian Organogenesis along the Cortical–Medullary Axis in Mammals"

_ijms, 2022, doi:10.3390/ijms232113373_

Round 1
Reviewer 1 Report
The article deals with an interesting subject of gonadal sex differentiation and ovarian organogenesis along the cortical-medullary axis in mammals.
It serves as an informational source for the mentioned theme, but is written in a very basic style, lacking more reports of unknown details in the field (not only for the ovary, also for sex differentiation), also critical thinking (e.g. comments on presented data). The sentences are often poorly connected in logical sense making this manuscript hard to read. Lots of comma is used so many sentences look like a listing of data. E.g "At E12.5, pregranulosa cells (Pre-supp.) in the medullary 843 region (green cells), which are the counterparts of Sertoli cells and primary sex cords, express an early ovarian factor, FOXL2 (red). 844 In contrast to testis differentiation, the proliferation and ingression of the CE in XX gonads continues after E12.5, leading to the 845 formation of secondary sex cords in the cortical region (orange cells)."
There are numerous grammar errors in the sentences, where some of them are marked in the pdf. These sentences, along with the whole manuscript should be revised.
The figures are very basically made. I advise to use Biorender to make them more attractive if possible. It is not clearly visible which type of cells are secreting which factor. Please fix this or provide clear information of this in the figure legends. For example, it is said in the figure legend that "Sox9 expression induces FGF9 and prostaglandin D2 (PGD2) production, forming positive-feedback loops to maintain their expression" - by using this style of writing, it is unclear which cell is an inducer and which is a responder.
Some of the terms shown on the figure and in figure legends are not reported in the manuscript text, like Pre-supp, or E-granulosa, or Pre-soma.
Also, on the figures, for some cells, (e.g.PGC) the term cell is listed, and for some not (Fetal Leydig, Sertoli, Pre-soma). This should be fixed.

Author Response
Response to Reviewer 1 Comments
To Reviewer: 1
We appreciate your reviewing our manuscript entitled " Gonadal sex differentiation and ovarian organogenesis along the cortical–-medullary axis in mammals". Your comments with a keen point of view help the paper to be better. We revised the paper according to the reviewer’s comments. The changes are highlighted by the track changes mode in MS Word. The specific action to your comment is also listed below. Your kind disposition of the paper would be greatly appreciated. Please find the revised manuscript/figures and lists of our actions to your comments below.
Thank you for your time and consideration.
Sincerely yours,
Lists of Actions to Comments
Comment #1:
The article deals with an interesting subject of gonadal sex differentiation and ovarian organogenesis along the cortical-medullary axis in mammals.
It serves as an informational source for the mentioned theme, but is written in a very basic style, lacking more reports of unknown details in the field (not only for the ovary, also for sex differentiation), also critical thinking (e.g. comments on presented data). The sentences are often poorly connected in logical sense making this manuscript hard to read. Lots of comma is used so many sentences look like a listing of data. E.g "At E12.5, pregranulosa cells (Pre-supp.) in the medullary 843 region (green cells), which are the counterparts of Sertoli cells and primary sex cords, express an early ovarian factor, FOXL2 (red). 844 In contrast to testis differentiation, the proliferation and ingression of the CE in XX gonads continues after E12.5, leading to the 845 formation of secondary sex cords in the cortical region (orange cells)."
Response #1:
To respond to your and the other reviewer’s comments, we rewrite and discussed the comments of the data and the future perspective (ex. lines. 321-332, lines 509-512, lines 529-533, lines 549-585). Especially, we raised the point to investigate this theme; the mechanism of how the formation of the secondary sex cords is occurred only in the XX gonads, not in the XY gonads. Furthermore, we modified the chapter “2. Molecular and Cellular Events in Ovarian Somatic Cells” following the other reviewer’s comments to make the manuscript straight. Along the time course of ovarian development, the order of some descriptions was changed (ex. lines 226-231 from 255-260). In addition, the comparison of the roles of FOXL2 and related genes in ovarian development (lines 297-332) and the description of the example of sex differentiation in turtles as non-mammalian vertebrates (lines 494-512) were added. With these modifications, 7 references were newly added, and the order of some references was changed, although no reference was deleted from the previous manuscript.
In response to your kind indication, we rewrote and checked the whole manuscript. Primarily, we separated the long sentences using some commas as much as possible. Additionally, the manuscript has undergone English language editing by MDPI.
Comment #2:
There are numerous grammar errors in the sentences, where some of them are marked in the pdf. These sentences, along with the whole manuscript should be revised.
Response #2:
We are sorry that there are numerous grammar errors. In response to your kind indication, we revised and checked the whole manuscript. Additionally, the manuscript has undergone English language editing by MDPI. Please see the attached certificate.
Comment #3:
The figures are very basically made. I advise to use Biorender to make them more attractive if possible. It is not clearly visible which type of cells are secreting which factor. Please fix this or provide clear information of this in the figure legends. For example, it is said in the figure legend that "Sox9 expression induces FGF9 and prostaglandin D2 (PGD2) production, forming positive-feedback loops to maintain their expression" - by using this style of writing, it is unclear which cell is an inducer and which is a responder.
Response #3:
To clear which type of cells are secreting the diffusible factors, such as FGF9 and PGD2, and which type of cells receive these factors, we modified Figs. 1B and C. In Fig. 1B, we show that FGF9 and PGD2 are secreted from Sox9-positive Sertoli cells and are received by the own and neighboring Sox9-positive Sertoli cells by using arrows. Additionally, we described “SOX9-positive Sertoli cells secrete FGF9 and PGD2, and these factors upregulate and maintain Sox9 expression in the own and neighboring supporting cell precursors.” in figure legends.
Comment #4:
Some of the terms shown on the figure and in figure legends are not reported in the manuscript text, like Pre-supp, or E-granulosa, or Pre-soma.
Also, on the figures, for some cells, (e.g.PGC) the term cell is listed, and for some not (Fetal Leydig, Sertoli, Pre-soma). This should be fixed.
Response #4:
In response to this comment, we deleted the terms not reported in the manuscript text. Additionally, we modified the terms like Pre-supp, or E-granulosa, or Pre-soma to the terms like supporting cell precursor, granulosa cells (shown as orange cells in Fig. 2C), and somatic cell precursor, respectively. Furthermore, we listed the terms of all drawing types of cells on the right side of the Figures.

Reviewer 2 Report
Very interesting review! Authors presented the recent findings about molecular and cellular events in gonadal sexual differentiation and ovarian organogenesis. Related studies were well cited and all these findings were organized well to make the story interesting and convincing. Just minor modification is needed to make this review better.
1. Please carefully check all mentioned genes. Are you talking about human or mice and correct the upper/lower letters of the gene name.
2. Authors described important genes in sexual differentiation in multiple species to emphasize the importance. However, it will be better if authors could compare the similarity and difference in significant time points/mechanism especially the ovarian organogenesis, which makes the function clearer.
3. About non-mammalian vertebrates, more studies could be reviewed. For example, some studies about sex determination in fish were published and could be a good resource in this review.
4. Some sentences are too long/complicated to understand well for first time reading. Authors could re-write in a simpler/more straightforward way to make it easier to read and understand.
Author Response
Response to Reviewer 2 Comments
To Reviewer 2
We appreciate your reviewing our manuscript entitled " Gonadal sex differentiation and ovarian organogenesis along the cortical–-medullary axis in mammals". Your kind comments helped the paper to be improved. We revised the paper according to the reviewer’s comments. The changes are highlighted by the track changes mode in MS Word. The specific action to your comment is also listed below. Your kind disposition of the paper would be greatly appreciated. Please find the revised manuscript/figures and lists of our actions to your comments below
Thank you for your time and consideration.
Sincerely yours,
Lists of Actions to Comments
Comment #1
Please carefully check all mentioned genes. Are you talking about human or mice and correct the upper/lower letters of the gene name.
Response #1:
In response to this comment, we checked all gene names in the whole manuscript and corrected the style or the upper/lower letters of the gene name.
Comment #2
Authors described important genes in sexual differentiation in multiple species to emphasize the importance. However, it will be better if authors could compare the similarity and difference in significant time points/mechanism especially the ovarian organogenesis, which makes the function clearer.
Response #2:
Following your kind advice, we modified the chapter “2. Molecular and Cellular Events in Ovarian Somatic Cells”. Along the time course of ovarian development, the order of some descriptions was changed (ex. lines 226-231 from 255-260). Additionally, we added the description of RUNX1 and its relationship with FOXL2 (lines 297-310). Consequently, we discuss the comparison of FOXL2 and related genes/signaling between mice, goats, and humans (lines 321-332). With these modifications, three references were newly added, and the order of some references was changed, although no reference was deleted from the previous manuscript.
Comment #3
About non-mammalian vertebrates, more studies could be reviewed. For example, some studies about sex determination in fish were published and could be a good resource in this review.
Response #3:
Following your kind advice, we added the description regarding sex differentiation in turtles (lines 494-512). With these modifications, 4 references were newly added. By this addition, we could show that the regionalization of the gonad along the cortical–medullary axis is conserved among various vertebrate species, even when different mechanisms of gonadal sex differentiation are adopted. In contrast, we considered adding the description regarding sex differentiation in fish, but the regionalization along the cortical–medullary axis is unclear. Therefore, we are sorry, but the studies about sex determination in fish were not included in this review.
Comment #4
Some sentences are too long/complicated to understand well for first time reading. Authors could re-write in a simpler/more straightforward way to make it easier to read and understand.
Response #4:
We are sorry that there are too long/complicated sentences. We rewrote and checked the whole manuscript in response to your kind indication. Primarily, we separated the long sentences using some commas as much as possible. Additionally, the manuscript has undergone English language editing by MDPI. Please see the attached certificate.

Round 2
Reviewer 1 Report
The article is now acceptable as the authors have included the necessary changes.